# RING finger E3 ligase PPP1R11 regulates TLR2 signaling and innate immunity

**Alison C McKelvey**[1,2†], **Travis B Lear**[1,2,3,4†], **Sarah R Dunn**[1,2†], **John Evankovich**[1,2], **James D Londino**[1,2], **Joseph S Bednash**[1,2], **Yingze Zhang**[1,2], **Bryan J McVerry**[1,2], **Yuan Liu**[1,2], **Bill B Chen**[1,2,5]*

[1]Department of Medicine, University of Pittsburgh, Pittsburgh, United States; [2]Acute Lung Injury Center of Excellence, University of Pittsburgh, Pittsburgh, United States; [3]Department of Environmental and Occupational Health, University of Pittsburgh, Pittsburgh, United States; [4]School of Public Health, University of Pittsburgh, Pittsburgh, United States; [5]Vascular Medicine Institute, University of Pittsburgh, Pittsburgh, United States

**Abstract** Toll-like receptor 2 (TLR2) is a pattern recognition receptor that recognizes many types of PAMPs that originate from gram-positive bacteria. Here we describe a novel mechanism regulating TLR2 protein expression and subsequent cytokine release through the ubiquitination and degradation of the receptor in response to ligand stimulation. We show a new mechanism in which an uncharacterized RING finger E3 ligase, PPP1R11, directly ubiquitinates TLR2 both in vitro and in vivo, which leads to TLR2 degradation and disruption of the signaling cascade. Lentiviral gene transfer or knockdown of PPP1R11 in mouse lungs significantly affects lung inflammation and the clearance of *Staphylococcus aureus*. There is a negative correlation between PPP1R11 and TLR2 levels in white blood cell samples isolated from patients with *Staphylococcus aureus* infections. These results suggest that PPP1R11 plays an important role in regulating innate immunity and gram-positive bacterial clearance by functioning, in part, through the ubiquitination and degradation of TLR2.

*For correspondence: chenb@ upmc.edu

†These authors contributed equally to this work

**Competing interests:** The authors declare that no competing interests exist.

## Introduction

The human toll-like receptor (TLR) family consists of ten family members (TLR1-TLR10) (*Chuang and Ulevitch, 2004*). TLRs are single transmembrane pattern recognition receptors that recognize molecules derived from various pathogens, referred to as pathogen-associated molecular patterns (PAMPs) (*Crespo-Lessmann et al., 2010*; *Delgado et al., 2008*; *Esen and Kielian, 2006*; *Chalifour et al., 2004*; *Tobian et al., 2003*) . TLRs are highly conserved from invertebrates to mammals, and are essential for mediating innate immunity and the production of cytokines in response to infectious agents (*O'Neill, 2000*). All TLRs share common structural features, including multiple leucine-rich repeats (LRR), a transmembrane domain, and a conserved cytoplasmic Toll−interleukin 1 receptor (IL-1R) domain (TIR domain) (*Chuang and Ulevitch, 2004*) . Specifically, LRR motifs from individual TLRs provide ligand binding sites for diverse PAMPs, whereas the TIR domain interacts with several intracellular proteins such as MyD88, which is essential for the transduction of downstream signaling (*Fitzgerald et al., 2001*; *Xu et al., 2000*). Within the TLR family, TLR2 possesses the unique ability to recognize glycolipids, lipopeptides, lipoproteins, and lipoteichoic acids from gram-positive bacteria. Thus, TLR2 is the key element of innate immunity that defends against gram-positive bacteria (*Raby et al., 2013*). TLR2 is expressed not only in immune cells, but is also present in pulmonary alveoli and airway epithelial cells, suggesting that it plays a role in mucosal innate immunity and infection-induced lung injury (*Charles et al., 2011*; *Hertz et al., 2003*;

*Hoth et al., 2012*; *Jiang et al., 2005*). Interestingly, TLR2 deficient mice are highly susceptible to *S. aureus* infection (*Takeuchi et al., 2000*), and in humans, a loss-of-function TLR2 mutation has been linked to susceptibility to infectious and inflammatory diseases, faster disease progression, and a more severe course of sepsis in critically ill patients (*Bronkhorst et al., 2013*; *Janardhanan et al., 2013*; *Nachtigall et al., 2014*; *Stappers et al., 2014*). These studies illustrate the protective role of TLR2 in response to inflammatory insults and infectious diseases. Therefore, TLR2 augmentation may be a novel therapeutic strategy in the fight against gram-positive infection.

Ubiquitination of proteins brands them for degradation, either via the proteasome or the lysosome, and regulates diverse processes (*Tanaka et al., 2008*). First, using ATP, the catalytically active cysteine (C) residue of the E1-activating enzyme forms a high energy thioester bond to the C-terminal glycine of ubiquitin (Ub) and then, using another ATP, transfers it to the active center C of the E2 Ub conjugating enzyme (Ubc) (*Jin et al., 2007*). Finally, the C-terminus of Ub is attached to the ε-amino lysine (K) residue of the substrate, mediated by an E3 ligase that typically targets a substrate marked by a particular post-translational degron (*Skaar et al., 2013*). The mechanism of protein ubiquitination is dependent on the type of E3 ligase: SCF, RING finger, U-box, or HECT (*Jin et al., 2007*; *Skaar et al., 2013*; *Hatakeyama et al., 2001*; *Dikic and Robertson, 2012*). Of the many E3 ligases, the RING finger domain E3 superfamily remains poorly characterized. Functional data are available for only about a dozen of the over 300 predicted ligases, such as MDM2 and several RNF and TRIM proteins (*Fang et al., 2000*; *Joazeiro and Weissman, 2000*; *Lorick et al., 1999*; *Metzger et al., 2012*). RING finger E3 ligases contain a unique RING (Really Interesting New Gene) finger domain that consists of two zinc finger type domains (*Borden and Freemont, 1996*; *Freemont et al., 1991*; *Lovering et al., 1993*). In this study, we characterize a previously undescribed RING finger E3 ligase family member, PPP1R11, and identify its role in regulating cytokine secretion through targeted TLR2 protein ubiquitination and degradation. These studies describe a new molecular mechanism contributing to mucosal innate immunity.

## Results

### TLR2 polyubiquitination is regulated by RING E3 ligase PPP1R11

We first investigated TLR2 protein degradation using a transformed murine lung epithelial (MLE) cell line. MLE cells express high levels of TLR2 protein (Data not shown). Cells were stimulated with the TLR2 ligand, Pam3CSK4, and TLR2 protein levels decreased after 4 hr. However, co-treatment with the proteasomal inhibitor MG-132 preserved TLR2 protein levels, while treatment with the lysosomal inhibitor leupeptin did not. Further, cell lysates from these treatments were subjected to TLR2 immunoprecipitation and Ubiquitin immunoblotting. High molecular weight ubiquitin signals in TLR2 immunoprecipitate were detected only in cell lysates treated with MG-132 (*Figure 1A*). Taken together, this suggests that the TLR2 ligand, Pam3CSK4, induces the polyubiquitination and subsequent degradation of TLR2 via the proteasome. We also investigated the type of ubiquitin linkage on TLR2 using a UbiCRest (Ubiquitin Chain Restriction Analysis) assay (*Hospenthal et al., 2015*). Polyubiquitinated TLR2 was subjected to several deubiquitinating enzymes (DUB) that target specific ubiquitin linkages (*Figure 1—figure supplement 1A*). As shown in *Figure 1B*, USP2, a DUB that non-specifically targets all ubiquitin linkages, completely reduced polyubiquitinated TLR2. We also observed a drastic decrease in TLR2 ubiquitination in the OTUB1 treated sample, which suggests that TLR2 ubiquitination is regulated in part through K48 ubiquitin linkage. We also observed K48-specific ubiquitin linkage of TLR2 following TLR2 immunoprecipitation and immunoblotting (*Figure 1—figure supplement 1B*). MG-132 dramatically increased the half-life of TLR2 (*Figure 1—figure supplement 1C*) and Ubiquitin overexpression drastically reduced TLR2 protein levels and half-life (*Figure 1—figure supplement 1D*, *Figure 1C*). These studies suggest that TLR2 protein degradation occurs in a ubiquitin-dependent manner.

Next, we tested over fifty E3 ligases that might be involved in TLR2 degradation (Data not shown) and found that overexpression of PPP1R11 decreased TLR2 levels (*Figure 1—figure supplement 2A*). Ectopic expression of PPP1R11 triggered TLR2 degradation in a dose dependent manner (*Figure 1D*) without affecting its mRNA levels (*Figure 1—figure supplement 2B*). Furthermore, ectopic PPP1R11 expression significantly decreased the half-life of TLR2 (*Figure 1E*), whereas PPP1R11 knockdown markedly increased its lifespan (*Figure 1F*, *Figure 1—figure supplement 2C*).

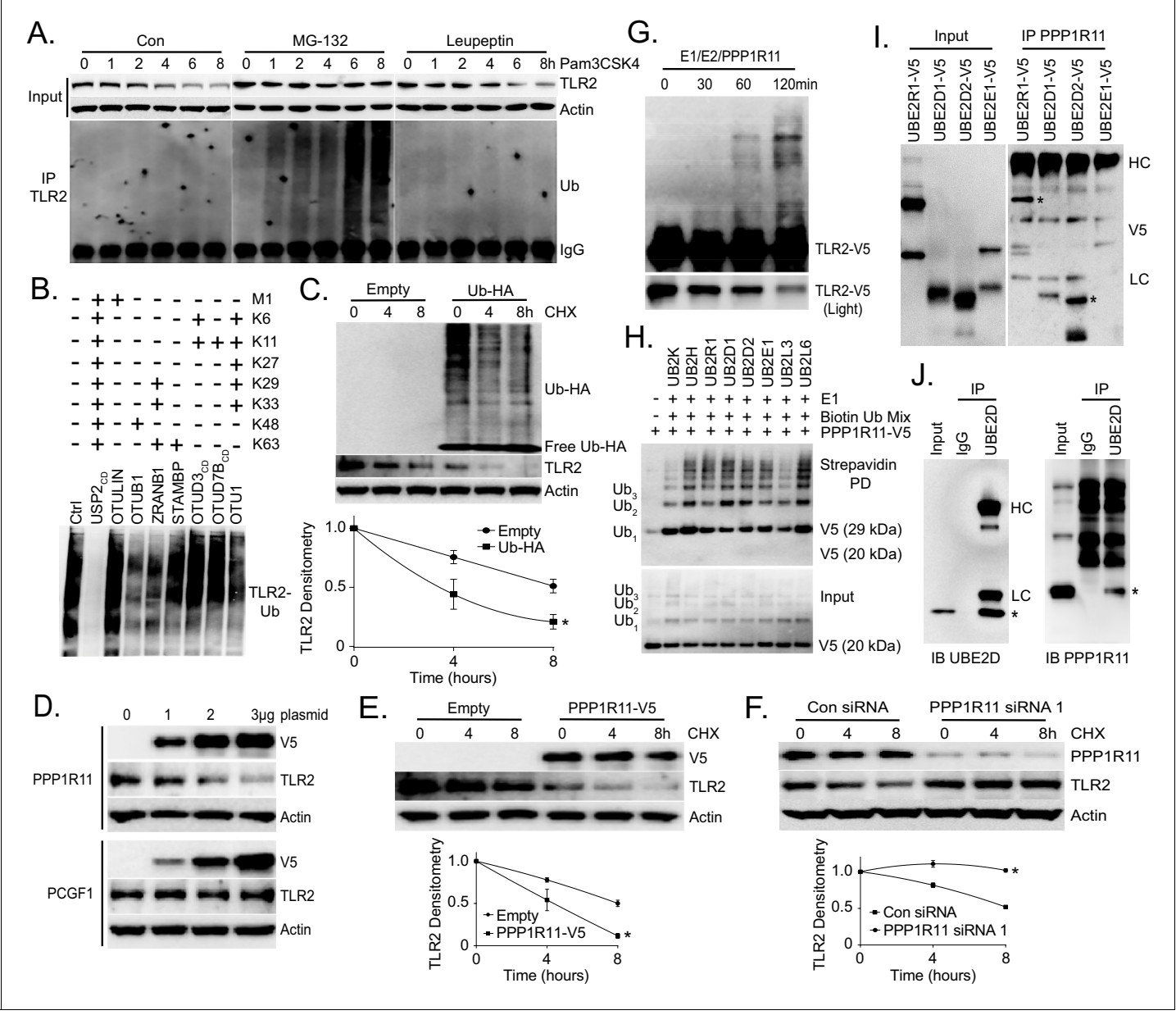

**Figure 1.** TLR2 polyubiquitination is regulated by PPP1R11. (**A**) Murine lung epithelial (MLE) cells were treated with Pam3CSK4 with or without MG-132 or Leupeptin in a time-dependent manner. Cells were collected and immunoblotted for TLR2 and Actin. Endogenous TLR2 was also immunoprecipitated and followed by Ubiquitin immunoblotting. (**B**) MLE cells were pretreated with MG-132 for 18 hr to induce accumulation of polyubiquitinated TLR2 protein. Endogenous TLR2 was immunoprecipitated using TLR2 antibody and protein A/G beads. After washing, TLR2 beads were then incubated with different DUBs before being assayed for ubiquitin immunoblotting. (**C**) TLR2 protein half-life determination in MLE cells transfected with Empty plasmid or Ubiquitin plasmid. Cells were collected and immunoblotted for TLR2, HA, and Actin. Below the panel levels of each protein on immunoblots were quantified densitometrically (Normalized to time zero) and shown graphically. The data represent mean values ± SEM (n = 3); *, p<0.05, significant compared to Control, Student's *t*-test. (**D**) Immunoblots showing levels of TLR2 proteins, V5, and Actin after PPP1R11 or PCGF1 plasmid dose over-expression. (**E–F**) TLR2 protein half-life determination in MLE cells with empty or PPP1R11 plasmid expression (**E**); TLR2 protein half-life determination with Control or *PPP1R11 siRNA* expression (**F**). Below each panel levels of each protein on immunoblots were quantified densitometrically (Normalized to time zero) and shown graphically. The data represent mean values ± SEM (n = 3); *p<0.05, significant compared to Control, Student's *t*-test. (**G**) In vitro ubiquitination assay. Purified E1 and E2 components were incubated with TLR2-V5, PPP1R11, and the full complement of ubiquitination reaction components for different lengths of time. (**H**) In vitro ubiquitination assay. PPP1R11-V5 was incubated with purified E1, varying E2s, biotinylated ubiquitin, and the full complement of ubiquitination reaction components. The reaction mixture was subjected to streptavidin pulldown prior to V5 immunoblotting. (**I**) PPP1R11 protein was immunoprecipitated from cell lysate using a PPP1R11 antibody and coupled to protein A/G beads. PPP1R11 beads were then incubated with in vitrosynthesized products expressing V5-tagged E2 conjugating enzymes. After

*Figure 1 continued on next page*

*Figure 1 continued*

washing, proteins were eluted and processed for V5 immunoblotting. (J) Endogenous PPP1R11 was also immunoprecipitated and immunoblotted for UBE2D.

The following source data and figure supplements are available for figure 1:

**Source data 1.** This file contains raw source data used to create graphs in *Figure 1*.

**Figure supplement 1.** TLR2 is regulated by Ubiquitination.

**Figure supplement 1—source data 1.** This file contains raw source data used to create graphs in *Figure 1—figure supplement 1*.

**Figure supplement 2.** The RING E3 ligase PPP1R11 regulates TLR2 stability.

**Figure supplement 2—source data 1.** This file contains raw source data used to create graphs in *Figure 1—figure supplement 2*.

**Figure supplement 3.** PPP1R11 is a non-canonical RING E3 ligase.

**Figure supplement 4.** PPP1R11 does not regulate the stability of other TLR receptors.

Conversely, we observed no effect on TLR2 decay with the expression and silencing of fellow RING E3 ligase PCGF1 (*Figure 1—figure supplement 2D–E*). To confirm PPP1R11 as an authentic TLR2 target, we performed an in vitro ubiquitination assay using TLR2 as the substrate and observed poly-ubiquitination of TLR2 (*Figure 1G*).

To investigate the in vivo relevance of the PPP1R11/TLR2 pathway, we assayed white blood cell (WBC) pellets for PPP1R11 and TLR2 expression in control and *S. aureus*-infected patients from our Acute Lung Injury Registry and Biospecimen Repository (*Coon et al., 2015*). Control patients were mechanically ventilated patients intubated for airway protection without clinical evidence of or risk factors for ARDS (Acute Respiratory Distress Syndrome), while *S. aureus*-infected patients had evidence of *S. aureus* in tracheal aspirates, blood, or both, and had evidence of or risk factors for ARDS. In control patients, we observed no correlation between PPP1R11 and TLR2 levels, but observed a significantly negative PPP1R11/TLR2 correlation in *S. aureus*-infected patients. We interpret these data to suggest that during *S. aureus* infection, TLR2 levels may be attenuated in part by PPP1R11 expression. This study suggests that the PPP1R11/TLR2 pathway may be relevant in the innate immune response to *S. aureus* infection in patients (*Figure 1—figure supplement 2F*).

RING finger E3 ligases utilize several unique residues, such as cysteine and histidine, to interact with zinc and form the two RING finger domains required for E3 ligase activity (*Borden, 1998*). They are characterized by the unique property of auto-ubiquitination (*Amemiya et al., 2008*; *Deshaies and Joazeiro, 2009*). We confirmed that PPP1R11 is an authentic ring finger E3 ligase by showing its ability to auto-ubiquitinate in the presence of ubiquitin, E1, and several different E2 conjugating enzymes (*Figure 1H*). We also showed that several E2 conjugating enzymes such as UBE2R1 and UBE2D2 preferably interact with PPP1R11 in vitro (*Figure 1I*). Lastly, we showed that PPP1R11 interacts with endogenous UBE2D2 through cellular co-immunoprecipitation (*Figure 1J*). We also selectively mutated several key residues of PPP1R11 putative RING finger (*Figure 1—figure supplement 3A*) and showed their loss-of-function in inducing TLR2 degradation (*Figure 1—figure supplement 3B*). H126 was chosen as negative control since it is outside of the RING finger domains. The effect of PPP1R11 was specific to TLR2, as an ectopic expression of PPP1R11 did not alter TLR3, TLR7, TLR8, or TLR9 protein levels (*Figure 1—figure supplement 4*). These experiments suggest that PPP1R11 is an authentic RING finger E3 ligase that specifically induces TLR2 ubiquitination and degradation.

## PPP1R11 regulates Pam3CSK4-induced TLR2 protein degradation and inflammation

To further elucidate the PPP1R11/TLR2 pathway in cells, we treated MLE cells with the TLR2 ligand, Pam3CSK4. We observed a time-dependent increase in PPP1R11 protein levels and decrease in

TLR2 protein levels with this treatment (*Figure 2A*). Further, cell lysates from these conditions were subjected to TLR2 immunoprecipitation, and high levels of PPP1R11 protein were detected in cells treated with Pam3CSK4 for 4 and 6 hr. PPP1R11/TLR2 association at these time points also reflects the highest degree of TLR2 protein loss (*Figure 2A*). We also examined the role of PPP1R11 in Pam3CSK4-induced TLR2 protein degradation, and upon ectopic expression of PPP1R11, we observed an accelerated decay of TLR2 protein with Pam3CSK4 treatment (*Figure 2B*). Last, we performed PPP1R11 knockdown in MLE cells, and we observed stabilized TLR2 protein levels as well as less ubiquitination upon Pam3CSK4 treatment as compared to control siRNA (*Figure 2C*). TLR2 immunoprecipitation also showed protein ubiquitination beginning at 1 hr post Pam3CSK4 treatment, which is consistent with increased PPP1R11/TLR2 association (*Figure 2A*). These studies suggest that the TLR2 ligand, Pam3CSK4, induces TLR2 degradation, in part, through the ubiquitination of TLR2 by the E3 ligase PPP1R11.

We also generated a series of TLR2 lysine mutants to further characterize the mechanism of ubiquitin transfer to TLR2. Mutation of lysine 754 to arginine (K754R) within TLR2 resulted in its highest level of stability, a much extended half-life (*Figure 2—figure supplement 1A*), and resistance to Pam3CSK4-induced degradation (*Figure 2—figure supplement 1B*). K754R TLR2 is also resistant to in vitro ubiquitination and degradation during PPP1R11 co-expression (*Figure 2—figure supplement 1C–D*). We confirmed that the K754R mutant TLR2 is expressed and localized normally in cells (*Figure 2—figure supplement 1E*). This suggests that the extended stability of the TLR2 K754R mutant is due to its inability to be ubiquitinated by PPP1R11 and that K754 is a potential PPP1R11 ubiquitination site within TLR2. Since Pam3CSK4 induces the TLR2 signaling pathway that leads to cytokine release, we hypothesized that by ubiquitinating TLR2, the E3 ligase PPP1R11 is able to suppress the TLR2 signaling pathway. Indeed, ectopic expression of PPP1R11 reduces both IL-6 and CXCL1 cytokines by 40–50% upon Pam3CSK4 treatment (*Figure 2D–E*), whereas PPP1R11 knockdown significantly increased both IL-6 and CXCL1 cytokine release in MLE cells (*Figure 2F–G*). Cytokine release upon overexpression and knockdown of fellow RING E3 ligase Trim52 resulted in no significant difference relative to controls (*Figure 2—figure supplement 1F–G*). Finally, we embarked on gene editing experiments to confirm the regulatory role of PPP1R11 on TLR2 and cytokine release. Utilizing CRISPR-Cas9 technology, we generated a 2 bp deletion in the first exon of *Ppp1r11* in MLE cells leading to an immediate pre-mature stop codon. Following colony expansion, we assayed protein half-life and observed stabilized TLR2 half-life in the *Ppp1r11* KO cells relative to control (*Figure 2—figure supplement 2A*). We also challenged PPP1R11 KO cells with PAM3CSK4, and observed increased IL-6 and CXCL1 cytokine release relative to control cells (*Figure 2—figure supplement 2B–C*). From these observations we believe PPP1R11 suppresses TLR2-linked inflammatory signaling.

## PPP1R11 gene transfer reduces lung inflammation and decreases bacterial clearance

To further characterize the role of PPP1R11 in TLR2-mediated infection, mice were infected with an empty lentivirus or lentivirus encoding PPP1R11. The mice were then challenged with *Staphylococcus aureus* (intratracheally, $10^8$ CFU). *S. aureus*, a gram-positive bacterium, is a significant contributor to nosocomial pneumonia, sepsis-associated acute lung injury (ALI), and acute respiratory distress syndrome (ARDS) (*Hudson et al., 1995*; *Ware and Matthay, 2000*). Mice were euthanized before analysis of lung inflammation and systemic bacterial loads. PPP1R11 gene transfer significantly reduced lung inflammation shown by reduced lavage cytokines, protein concentrations, cell counts, and cell infiltrates (*Figure 3A–E, and L*). However, there were significantly higher bacterial loads in the lung, lavage fluid, blood, and liver (*Figure 3F,G,H, and I*), and there was a trend towards an increase in the spleen and kidney (*Figure 3J,K*). We also showed that in mice infected with *S. aureus*, PPP1R11 expression in the lung effectively reduced TLR2 protein levels (*Figure 3M*).

## PPP1R11 knockdown induces lung inflammation and increases bacterial clearance

To further confirm that PPP1R11 suppresses an inflammatory response through TLR2 in vivo, mice were first infected with lentivirus encoding control shRNA or *Ppp1r11* shRNA and then challenged with *S. aureus* (intratracheally, 2.5*107 CFU). PPP1R11 knockdown significantly increased lavage

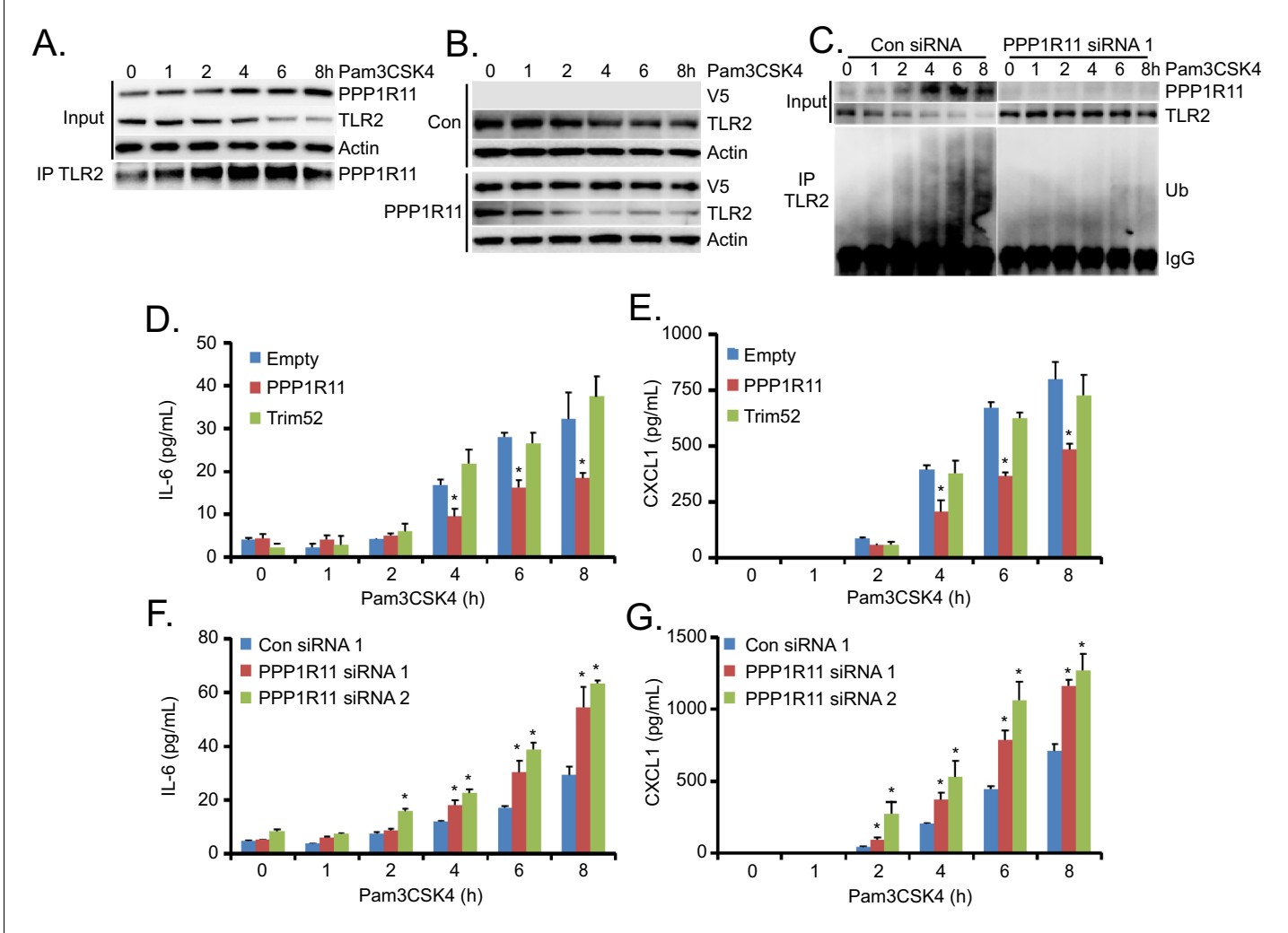

**Figure 2.** PPP1R11 regulates Pam3CSK4-induced TLR2 protein degradation and inflammation.  (A) MLE cells were treated with Pam3CSK4 in a time-dependent manner. Cells were collected and immunoblotted for PPP1R11, TLR2, and Actin. Endogenous TLR2 was also immunoprecipitated and immunoblotted for PPP1R11. (B) MLE cells were transfected with either empty or PPP1R11 plasmid. 24 hr later, cells were exposed to Pam3CSK4 in a time-dependent manner. Cells were collected and immunoblotted for V5, TLR2, and Actin. (C) MLE cells were transfected with either control or PPP1R11 siRNA. 48 hr later, cells were exposed to Pam3CSK4 in a time-dependent manner. Cells were collected and immunoblotted for V5, TLR2, and Actin. Endogenous TLR2 was also immunoprecipitated and followed by ubiquitin immunoblotting. (D–E) MLE cells were transfected with either empty, PPP1R11, or TRIM52 plasmid. 24 hr later, cells were exposed to Pam3CSK4 in a time-dependent manner. Cell media was then collected and assayed for IL-6 and CXCL1 (Data represent mean values ± SEM n = 4; *p<0.05, significant compared to Control, Student's t-test). (F–G) MLE cells were transfected with either control or *PPP1R11 siRNA*. 48 hr later, cells were exposed to Pam3CSK4 in a time-dependent manner. Cell media was then collected and assayed for IL-6 and CXCL1 (Data represent mean values ± SEM n = 4; *p<0.05, significant compared to Control, Student's t-test).

The following source data and figure supplements are available for figure 2:

**Source data 1.** This file contains raw source data used to create graphs in *Figure 2*.
**Figure supplement 1.** TLR2 ubquitination is regulated through Lysine 754.
**Figure supplement 1—source data 1.** This file contains raw source data used to create graphs in *Figure 2—figure supplement 1*.
**Figure supplement 2.** *Ppp1r11* KO cells rescue TLR2 stability and function.
**Figure supplement 2—source data 1.** This file contains raw source data used to create graphs in *Figure 2—figure supplement 2*.

cytokines, protein concentrations, cell counts, and cell infiltrates (*Figure 4A–E, and J*). Interestingly, increased lung inflammation further cleared bacterial loads in the lavage fluids, lung, and blood (*Figure 4F,G, and H*), and there was a trend towards decreased bacterial loads in the liver (*Figure 4I*). We also showed that in mice infected with *S. aureus,* PPP1R11 knockdown effectively increased TLR2 protein levels in the lungs (*Figure 4K*). Taken together, these results suggest that PPP1R11 is a negative regulator of TLR2 signaling and inflammatory cytokine release, which mediates the clearance of *S. aureus* after acute infection.

## Discussion

Pattern recognition receptors play a critical role in mucosal immunity by sensing PAMPs from invading pathogens and initiating cellular responses to eliminate those pathogens. Here we show a novel mechanism by which the previously unrecognized RING finger E3 ligase, PPP1R11, attenuates TLR2 signaling in response to *S. aureus* infection by targeting TLR2 for proteasomal degradation. PPP1R11 (Protein Phosphatase 1 Regulatory Subunit 11) was originally described to interact with and inhibit protein phosphatase 1 activity (*Zhang et al., 1998*) . It also has another name, TcTex5 (T-Complex-Associated-Testis-Expressed 5), and it was implied that mutation of TcTex5 is associated with a mouse sperm motility abnormality in sterile t-haplotype mutant mice (*Han et al., 2008*; *Pilder et al., 2007*). However, its E3 ligase activity has never been described. Canonical RING finger domains contain the consensus sequence $C-X_2-C-X_{[9-39]}-C-X_{[1-3]}-H-X_{[2-3]}-C-X_2-C-X_{[4-48]}-C-X_2-C$, where both cysteine and histidine residues are involved in zinc coordination (*Borden and Freemont, 1996*). However, we believe that PPP1R11 has a non-canonical RING finger domain with all the characteristics of a Ring finger E3 ligase (*Figure 1G,H,I*, *Figure 1—figure supplement 3A*). We show the relative specificity of TLR2 targeting by PPP1R11, as none of the other TLRs tested were degraded by PPP1R11 expression (*Figure 1—figure supplement 4*). However, it is possible that other E3 ligases target TLR2 for ubiquitination as this phenomenon has been well described with other substrates, e.g. SCF E3 ligase subunits Fbxw1, Fbxw7, Fbxw8, Fbxl1, Fbxo4, and Fbxo31 are all capable of ubiquitinating cyclin D1 (*Skaar et al., 2009*). We do not rule out other RING finger E3 ligase or even other E3 ligase family members to potentially target TLR2 for ubiquitination, however, our data suggest that PPP1R11 targets TLR2 for proteasomal degradation via ubiquitination.

Targeted TLR2 degradation by PPP1R11 may act as a negative feedback mechanism in the lung to prevent a harmful, excessive inflammatory response that causes severe lung injury. We have observed similar function of other E3 ligase proteins such as Fbxl19 in acute lung injury and pneumonia (*Zhao et al., 2012*). Indeed, we showed that treatment of MLE cells with TLR2 ligand, Pam3CSK4, up-regulates PPP1R11 protein levels and gradually reduces TLR2 protein levels at about 4–6 hr (*Figure 2A*). We also showed that ectopic expression of PPP1R11 in MLE cells significantly reduced Pam3CSK4-induced cytokine release, and gene editing of PPP1R11 significantly increases ligand-induced cytokine release (*Figure 2D–E*; *Figure 2—figure supplement 2B–C*). This evidence suggests that PPP1R11 could be a natural inflammatory suppressor in cells, and this pathway could be utilized to target multiple types of stimuli-induced inflammation. While the targeted degradation of TLR2 may restrain excessive inflammatory responses, activation of this pathway is also critical for pathogen clearance. By manipulating levels of TLR2 in murine lungs through PPP1R11, we showed effects in downstream inflammation and *S. aureus* bacterial clearance (*Figures 3* and *4*). These results are in line with previous studies showing TLR2 deficient mice were more susceptible to *S. aureus* infection (*Takeuchi et al., 2000*). Specifically, higher inflammation in the lung might lead to lung injury, but also proved to be more efficient at clearing bacteria (*Figure 4*), whereas reduced inflammation in the lung is tissue-protective but fails to clear bacteria, which leads to systemic bacteremia (*Figure 3*). One approach to potentially control the balance between the beneficial effects of the inflammatory pathway and the harmful and excessive responses to pathogens is to manipulate PPP1R11. These results suggest that PPP1R11 inhibition might confer a protective phenotype against *S. aureus* pneumonia by augmenting TLR2 signaling, which is critical for bacterial clearance. As a proof-of-concept investigation, we observed a negative correlation between TLR2 and PPP1R11 protein levels in WBC pellets from *S. aureus*-infected patients. Interestingly, no correlation between PPP1R11 and TLR2 levels was observed in control patients (*Figure 1—figure supplement 2D*). This experiment suggests that the PPP1R11/TLR2 pathway is only active during *S. aureus* infection. It is likely that other regulators such as kinases are also involved in PPP1R11-driven TLR2

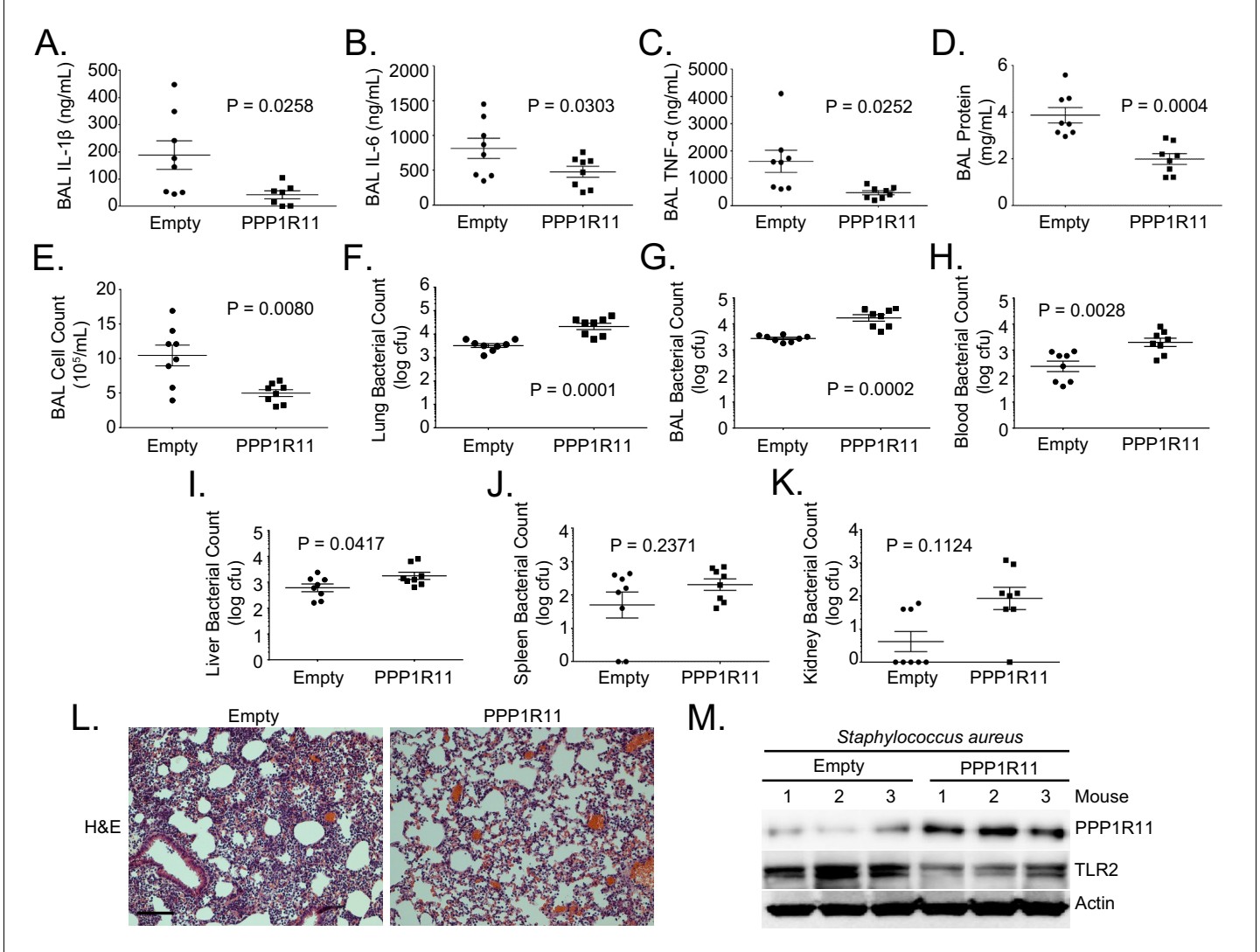

**Figure 3.** PPP1R11 gene transfer in the lung reduces lung inflammation and decreases bacterial clearance. Lenti-Empty or Lenti-PPP1R11 (107PFU/mouse) was administered i.t. to C57BL/6J mice for 144 hr; mice were then infected i.t. with *S. aureus* (108 CFU). Mice were euthanized after 18 hr, and lungs were lavaged with saline, harvested, and then homogenized. Blood, liver, spleen, and kidney were also harvested and homogenized. (**A–E**) and (**G**). Lavage cytokines, BAL protein, BAL cell counts, and BAL bacterial counts were measured. The data represent mean values ± SEM (n = 8 mice per group; p<0.05, significant compared to Control, Student's *t*-test). (**F**) and (**H–K**). Lung, blood, liver, spleen, and kidney bacterial counts were measured. The data represent mean values ± SEM (n = 8 mice per group; p<0.05, significant compared to Control, Student's *t*-test). (**L**) H and E staining was performed on lung samples. Original magnification, 20X. Bar indicates 100 μm. (**M**) Mice lungs were isolated and assayed for TLR2, PPP1R11, and Actin immunoblotting.

The following source data is available for figure 3:

**Source data 1.** This file contains raw source data used to create graphs in *Figure 3*.

ubiquitination, which is common in E3 ligase substrate targeting (*Chen et al., 2013*; *Lear et al., 2016*). Further studies are warranted to investigate regulators that influence PPP1R11 expression and activity.

In conclusion, we show that the previously uncharacterized RING E3 Ligase, PPP1R11, is up-regulated in response to the TLR2 ligand, Pam3CSK4, and subsequently targets TLR2 for proteasomal degradation by ubiquitinating it at lysine 754. Down-regulation of TLR2 levels by PPP1R11 reduces pro-inflammatory cytokine secretion but impairs bacterial clearance in response to *S. aureus*. Taken

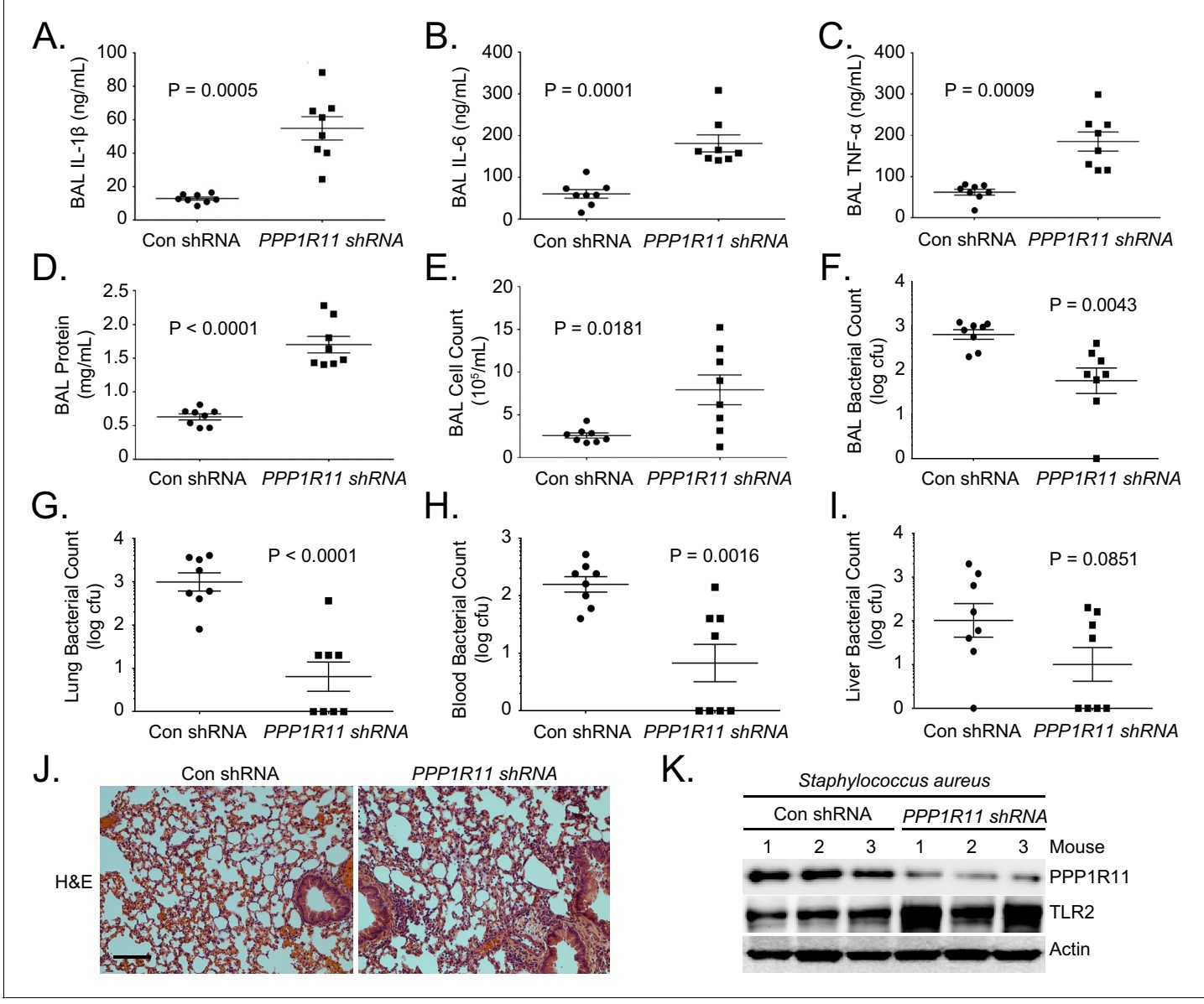

**Figure 4.** PPP1R11 knockdown induces lung inflammation and increases bacterial clearance. Lenti-control shRNA or Lenti-*PPP1R11* shRNA (107 PFU/ mouse) was administered i.t. to C57BL/6J mice for 144 hr; mice were then infected i.t. with *S. aureus* (2.5*10⁷ CFU). Mice were euthanized after 18 hr, and lungs were lavaged with saline, harvested, and then homogenized. Blood, liver, spleen, and kidney were also harvested and homogenized. (A–F) Lavage cytokines, BAL protein, BAL cell counts, and BAL bacterial counts were measured. The data represent mean values ± SEM (n = 8 mice per group; p<0.05, significant compared to Control, Student's *t*-test). (G–I). Lung, blood, and liver bacterial counts were measured. The data represent mean values ± SEM (n = 8 mice per group; p<0.05, significant compared to Control, Student's *t*-test).(J) H and E staining was performed on lung samples. Original magnification, 20X. Bar indicates 100 μm. (K) Mice lungs were isolated and assayed for TLR2, PPP1R11, and Actin immunoblotting.

The following source data is available for figure 4:

**Source data 1.** This file contains raw source data used to create graphs in *Figure 4*.

together, this study characterizes a novel innate immune regulatory mechanism involving PPP1R11-targeted degradation of the pattern recognition receptor TLR2.

## Materials and methods

### Materials

Sources of the murine lung epithelial (MLE) cell line were described previously (*Chen and Mallampalli, 2007*; *Ray et al., 2010*). QuikChange II XL Site-Directed Mutagenesis Kit (200522) was from Aglient Technologies. Nucleofector 2b and nucleofection kits (AAB-1001) were from Amaxa. High capacity RNA-to-cDNA kits (4387406) were from Applied Biosystems. HEK293T cells (CRL-3216) were from ATCC. Thermal Cycler Life ECO (BYQ6078) was from BIOER Technology. Secondary antibodies (170–515/6) and CFX96 Touch Real-Time qPCR (1855196) were from BioRad. UbiCRest assay kit (K-400) was from Boston Biochem. Anti-TLR2 (12276), anti-Ubiquitin (3936, RRID:AB_331292), and anti-HA (3724, RRID:AB_1549585) antibodies were from Cell Signaling. Lenti-X Packaging System (631276) was from Clontech. Murine IL-1b (88–7261) and murine IL-6 (88–7064, RRID:AB_2574986) ELISA kits were from eBioscience. Ubiquitin E1, E2s, biotinylated ubiquitin (BML-UW0400) and cycloheximide (BML-GR310) were from Enzo. FBS (100–106) was from Gemini. DNA sequencing was performed at Genewiz. Dulbecco's Modified Eagle Medium-F12 (11965–092) was from Gibco. Anti-V5 antibody (R960, RRID:AB_159298) was from Invitrogen. PAM3CSK4 (tlrl-pms) was from InvivoGen. C57BL/6J mice (000664, RRID:IMSR_JAX:000664) were from the Jackson Laboratory. SYBR Select Master Mix (4472918) was from Life Technologies. 35 mm Glass Bottom MakTek dishes (P35GCOL-0–10-C) were from MatTek. Alexa Fluor 488 Phalloidin (A12379, RRID:AB_2315147), Hoechst 33342 (H3570), Alexa Fluor 488 conjugate Donkey anti-Mouse IgG (R3711), Alexa Fluor 568 conjugate Goat anti-Mouse IgG (H+L) (A-11004), Normal Goat Serum (50197Z) were from Molecular Probes. TnTT7 Quick Coupled Transcription/Translation System (L1170) was from Promega. Murine CXCL1 (DY453) and Murine TNF-a (DY410) ELISA kits were from R and D Systems. PPP1R11 CRISPR/Cas9 KO plasmid (SC-429347), Anti PPP1R11 antibody (SC-135427, RRID:AB_10840420), Anti UBE2D (SC-166278, RRID:AB_2210152), Anti UBE2H (SC-100620, RRID:AB_2210469) were from Santa Cruz Biotechnology. Anti Actin antibody (A5441, RRID:AB_476744), leupeptin (L2884), TSB (22092), and Agar (A5306) were from Sigma Aldrich. PureLink RNA Mini kit (12183020), Strepavidin agarose resin (20349), and pcDNA3.1D V5/HIS/TOPO kit (K490040) were from Thermo Fisher Scientific. MG-132 (F1100) was from UBPBio.

### Cell culture

Murine Lung Epithelial 12 cells (MLE) were from ATCC (CRL-2110) and cultured according to manufacturer's instructions. The identity of the cell lines was monitored by microscope based morphology analysis and immunoblotting with multiple markers. The cell lines were checked for mycoplasma contamination using the MycoAlert Mycoplasma Detection Kit (Lonza, Switzerland). MLE cells were cultured in Dulbecco's Modified Eagle Medium-F12 (Gibco) supplemented with 10% fetal bovine serum (DMEM-F12-10). For PPP1R11 overexpression in MLE cells, an Amaxa nucleofection kit was used following the manufacturer's protocol. 24 hr later, cells were treated with doses of Pam3CSK4 up to 10 μg/ml. For PPP1R11 knockdown studies in MLE cells, scramble siRNA and PPP1R11 siRNA were used to transfect cells for 48 hr using electroporation. Cell lysates were prepared by brief sonication in 150 mM NaCl, 50 mM Tris, 1.0 mM EDTA, 2 mM dithiothreitol, 0.025% sodium azide, and 1 mM phenylmethylsulfonyl fluoride (Buffer A) at 4°C. For half-life study, MLE cells were exposed to cycloheximide (40 mg/ml) in a time dependent manner for up to 6 hr. Cells were then collected and immunoblotted. Protein densitometry was quantified through ImageJ and normalized to the zero time point for each set of condition.

### UbiCRest assay

TLR2 protein was analyzed for ubiquitin linkage specificity via the UbiCRest method and following the manufacturer's protocol (*Hospenthal et al., 2015*). MLE cells were treated with MG-132 (20 μM) for 18 hr prior to immunoprecipitation of endogenous TLR2 protein. TLR2 bound antibodies were conjugated to Protein A/G agarose resin and distributed among nine tubes. The TLR2-resins were incubated with deubiquitinases with varying specificities to the eight ubiquitin linkages. The deubiqutinases used and their working concentrations were USP2 (0.5 μM), OTULIN (0.5 μM), OTUB1 (0.5 μM), ZRANB1 (0.5 μM), STAMBP (0.5 μM), OTUD3 (0.5 μM), OTUD7A (0.1 μM), and OTU1 (0.5 μM).

Following the reaction, TLR2 protein was eluted from resin and subjected to ubiquitin immunoblotting.

## In vitro ubiquitin conjugation assays

The assay was performed in a volume of 25 ml containing 50 mM Tris pH 7.6, 5 mM MgCl$_2$, 0.6 mM DTT, 2 mM ATP, 1.5 ng/ml E1, 10 ng/ml Ubiquitin E2 conjugating enzymes, 1 mg/ml ubiquitin (Calbiochem), 1 mM ubiquitin aldehyde, and in vitro synthesized V5-TLR2 and PPP1R11. Reaction products were immunoblotted for V5.

## In vitro protein binding assays

PPP1R11 protein was immunoprecipitated from 1 mg MLE cell lysate using PPP1R11 antibody (rabbit) and coupled to protein A/G agarose resin. PPP1R11 beads were then incubated with in vitro synthesized products (50 ul) expressing V5-E2 conjugating enzymes. After washing, the proteins were eluted and processed for V5 immunoblotting.

## RT–qPCR, cloning, and mutagenesis

Total RNA was isolated and reverse transcription was performed followed by real-time quantitative PCR with SYBR Green qPCR mixture as described (*Butler and Mallampalli, 2010*). All mutant PPP1R11 and TLR plasmid constructs were generated using PCR-based approaches using appropriate primers and subcloned into a pcDNA3.1D/V5-His vector.

## Immunostaining

MLE cells were seeded in 35 mm MatTek glass-bottom dishes before the plasmid transfection. Cells were washed with PBS and fixed with 4% paraformaldehyde for 20 min, then exposed to 2% BSA, 1:500 mouse V5 antibody and 1:1000 Alexa 568 nm chicken anti-mouse antibody sequentially for immunostaining. The nucleus was counterstained with DAPI and F-actin was counterstained with Alexa 488-Phalloidin. Immunofluorescent cell imaging was performed on a Nikon A1 confocal microscope using 405 nm, 488 nm, or 567 nm wavelengths. All experiments were done with a 60x oil differential interference contrast objective lens.

## Lentivirus construction

To generate lentivirus encoding PPP1R11, the Plvx-PPP1R11 plasmid was co-transfected with Lenti-X HTX packaging plasmids (Clontech) into 293FT cells following the manufacturer's instructions. 72 hr later, the virus was collected and concentrated using Lenti-X concentrator.

## Gene editing

CRISPR guide RNAs (gRNA) were generated by *Santa Cruz Biotechnologies* specific to the first exon of *Ppp1r11* (TTGTAGGACGCCGTCCTTTG). MLE cells were transfected with 5 ug of plasmid. Transfection efficiency was confirmed via GFP expression. Cells were diluted and seeding to single-cell concentrations and sequence confirmed prior to half-life and cytokine studies.

## Animal studies

All procedures were approved by the University of Pittsburgh Institutional Animal Care and Use Committee. For pneumonia studies, C57BL6 micewere deeply anesthetized using a ketamine/xylazine mixture,and the larynx was well visualized under a fiber opticlight source before endotracheal intubation with a 3/400 24 gaugeplastic catheter. 10$^7$ CFU of lentivirus encoding genes for PPP1R11 or *PPP1R11* shRNA was instilled i.t. for 144 hr before the administration of *S. aureus* (strain 29213, 2.5*107-108 CFU/mouse, i.t.) for 18 hr, after which animals were euthanized and assayed for BAL protein, cell count, cytokines, lung infiltrates, and tissue bacterial count (*Coon et al., 2015*; *Zhao et al., 2012*; *Chen et al., 2013*; *Mallampalli et al., 2013*)

## Human samples

This study was approved by the University of Pittsburgh Institutional Review Board. As part of an ongoing Acute Lung Injury Biospecimen Repository, blood was collected from critically ill, mechanically-ventilated patients admitted to the University of Pittsburgh Medical Center Presbyterian

Hospital Medical Intensive Care Unit. Whole blood was treated with red blood cell lysis solution, and leukocytes were washed once with PBS prior to pelleting. Groups were separated into 'control' and 'injury' cohorts based on clinical evidence of or risk factors for Acute Respiratory Distress Syndrome (ARDS) as determined by an expert clinical panel. Control patients were mechanically ventilated without clinical evidence of or risk factors for Acute Respiratory Distress Syndrome (ARDS), while the injury cohort had clinical evidence of or risk factor for ARDS. For this study, samples were analyzed from a group of randomly selected control cohort patients (without *S. aureus* infection), as well as from patients with evidence of *S. aureus* infection (BAL, blood, or both) in the injury cohort. The Biospecimen Repository provided de-identified samples to this laboratory, and cell pellets were assayed for TLR2 and PPP1R11 proteins by immunoblotting, quantified using ImageJ software, and graphed in GraphPad. All data acquisition and densitometry was performed blinded prior to revealing cohort groups.

## Statistical analysis

Statistical comparisons were performed with the mean ± standard error of the mean (SEM) for continuous variables. All data were statistically analyzed by unpaired 2 sample t-test with $p < 0.05$ indicative of statistical significance. All analyses were performed using GraphPad Prism 6.

## Acknowledgements

This work was supported by the National Institutes of Health R01 grant HL116472 and HL132862 (to BBC), P01 grant HL114453 (BJM, YZ and BBC), and a University of Pittsburgh Vascular Medicine Institute seed fund. We thank Shristi Rajbhandari for her contributions on early E3 ligase screening.

## Additional information

### Funding

| Funder | Grant reference number | Author |
| --- | --- | --- |
| National Institutes of Health | HL114453 | Yingze Zhang<br>Bryan J McVerry<br>Bill B Chen |
| National Heart, Lung, and Blood Institute | HL116472 | Bill B Chen |
| National Heart, Lung, and Blood Institute | HL132862 | Bill B Chen |

The funders had no role in study design, data collection and interpretation, or the decision to submit the work for publication.

### Author contributions

ACM, TBL, Acquisition of data, Analysis and interpretation of data, Drafting or revising the article; SRD, BJM, Acquisition of data, Analysis and interpretation of data; JE, Conception and design, Analysis and interpretation of data; JDL, YL, Conception and design, Contributed unpublished essential data or reagents; JSB, Analysis and interpretation of data, Contributed unpublished essential data or reagents; YZ, Acquisition of data, Contributed unpublished essential data or reagents; BBC, Conception and design, Acquisition of data, Analysis and interpretation of data, Drafting or revising the article

### Author ORCIDs

Travis B Lear, http://orcid.org/0000-0001-9156-0844
Bryan J McVerry, http://orcid.org/0000-0002-1175-4874
Bill B Chen, http://orcid.org/0000-0003-2695-5107

### Ethics

Human subjects: This study was approved by the University of Pittsburgh Institutional Review Board, and consent was obtained.

Animal experimentation: All animal experiments were approved by the University of Pittsburgh Institutional Animal Care and Use Committee (IACUC) under protocol (14023127). Mice were housed at University of Pittsburgh Animal Care Facility and maintained according to all federal and institutional animal care guidelines.

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
