## [Decision Letter]

Thank you for submitting your article "RING Finger E3 Ligase PPP1R11 Regulates TLR2 Signaling and Innate Immunity" for consideration by *eLife*. Your article has been reviewed by Shao-Cong Sun (Reviewer #1) and Zhengfan Jiang (Reviewer #2) and the evaluation has been overseen by a Reviewing Editor and Tadatsugu Taniguchi as the Senior Editor.

The reviewers have discussed the reviews with one another and the Reviewing Editor has drafted this decision to help you prepare a revised submission.

Summary:

The authors demonstrate that RING finger E3 ligase PPP1R11 can modulates polyubiquitination and degradation of TLR2 via proteasome, thus regulating TLR2-triggered the innate immunity. PPP1R11 is up regulated by TLR2 ligation and subsequently ubiquitinates TLR2 at lysine 754 for degradation, thereby suppressing cytokine production induced by TLR2/gram-positive bacterial infection. This work identifies a previously unreported RING finger E3 ligase in TLR2 regulation.

Essential revisions:

Further detailed investigation is needed for TLR2 polyubiquitination and degradation, and also in vitro experiments with CRISPR approach would be needed to improve the quality of this work. Another RING finger E3 ligase should be added as control to support the specificity of PPP1R11 in the regulation of TLR2.

Reviewer #1:

In this manuscript, authors present a series of studies suggesting that a novel RING finger E3 ligase, PPP1R11, regulates polyubiquitination and degradation of TLR2 via proteasome. PPP1R11 is up regulated in response to the TLR2 ligand, Pam3CSK4, and subsequently ubiquitinates TLR2 at lysine 754 for degradation, thereby suppressing cytokine production induced by TLR2 signaling pathway. Overexpression or knockdown of PPP1R11 in mouse affects lung inflammation and bacterial clearance in response to staphylococcus aureus infection. Together, they conclude that PPP1R11 targets TLR2 for ubiquitination and degradation in vitro and in vivo, which is involved in the regulation of innate immunity induced by gram-positive bacterial infection. These findings are significant, since they characterize a previously unreported RING finger E3 ligase and suggest a new mechanism of TLR2 regulation. However, the authors should consider the following comments to further strengthen the work.

Major Points:

1) In Figure 1, protein level of TLR2 is significantly lower in lane 4 than lane 1; however TLR2 densitometry figures showed identical TLR2 densitometry in these lanes. The authors should explain how the densitometry was calculated and normalized. It appears that the main difference is in the untreated lanes.

2) In Figure 2, it appears that TLR2 degradation started from 1h following Pam3CSK4 stimulation, whereas PPP1R11 binding did not occur until 4h. Pam3CSK-induced TLR2 ubiquitination also started at 1h post-stimulation (Figure 2). The authors should repeat this experiment to see whether this result is reproducible or due to experimental variations. If these data are reproducible, they would suggest the existence of another E3 that mediates the early stage of TLR2 ubiquitination. In such a case, the authors should discuss the results.

3) The mechanism underlying TLR2 ubiquitination was not investigated in depth. The authors should use chain-specific ubiquitin antibodies to examine whether TLR2 is conjugated with K48 ubiquitin chains or other chains (e.g. Figure 1 and Figure 2).

4) In Figure 1, the protein level of TLR2 is drastically reduced after incubating TLR2 with E1, E2 and PPP1R11 in in vitro ubiquitin conjugation assays. Is this due to ubiquitination or degradation in the in vitro system? This result is quite different from that of Figure 1, central panel (MG-132).

5) Figure 1 should indicate the amount of the individual DUBs and show a panel of DUB protein immunoblot.

6) In Figure 1—figure supplement 3, the authors find several key residues of PPP1R11 that are important for TLR2 degradation. It would be helpful if the authors the possible structural mechanisms and any rationales for testing these residues (e.g. are these conserved residues in certain domains?).

Reviewer #2:

TLR2 is an important pattern recognition receptor that senses the pathogen-associated molecular patterns (PAMPs), mainly those derived from gram-positive bacteria, to initiate downstream immune pathways and inflammation. As such, data on the regulation of TLR2 is important for our understanding of the immune response. The authors present data on the role of a novel RING finger E3 ligase PPP1R11 in regulating TLR2 abundance in both human and murine cell lines, as well as by gene-transfer in mice organs. Overall the manuscript is written in a clear logic and the experiments are well designed. The data presented are novel, and the regulatory effects for PPP1R11 on the pathogenesis of lung infection have been defined.

However, there are some major issues to be carefully considered before the manuscript is ready for publication on *eLife*.

First, most of the data in this manuscript are from ectopic expression or RNAi-mediated gene knockdown experiments. As gene-specific knockout using the CRISPR-Cas9 system is technically mature, data collected with those PPP1R11 knockout cell-lines or mice, if applicable, should undoubtedly make the conclusion more solid and reliable.

Second, the effects of PPP1R11 in TLR2-mediated infection are double faced. From the gene transfer experiments, we can see that the gain or loss of PPP1R11 results in a selection between a control of inflammation and bacteria clearance. This phenomenon, however, makes it a hard decision to manipulate this gene for an effective treatment of lung infection. The authors should carefully think over this issue and determine the physiological effects of PPP1R11.

Third, while the manuscript is well written, there are areas that would benefit from re-working in terms of English grammar and usage. These would make the manuscript flow better and, therefore, make it more pleasant for the reader. I would like to stress however that, in its current form, the manuscript is both readable and understandable.

Specific Points:

1) The control panel probing TLR2 from Figure 1 does not show a time-dependent decrease in the protein level of TLR2, for the TLR2 level at 4 hour treatment is higher than that at 2 hour treatment. This inconsistency should be corrected.

2) Figure 1, another RING finger E3 ligase should be included in these experiments side by side with PPP1R11 to show that the latter is specific for TLR2. Using PCGF1 as in Figure 1 is just fine.

3) Figure 2, an siRNA for another RING finger E3 ligase should be included in these experiments to show specificity. Using Trim52 as in Figure 2 is just fine.

4) Figure 3, although a general trend is shown that PPP1R11 expression results in the loss of TLR2, individual consistency should be paid attention to, as gene transfer Mouse 1 should not display more TLR2 than Mouse 5 considering their PPP1R11 expression levels.

5) Figure 4, similar to the point raised for Figure 3, and more consistent panels are preferred for PPP1R11 and TLR2 expressions, if applicable.

---

## [Author Response]

*[…] Reviewer #1:*

*[…] Major Points:*

*1) In Figure 1, protein level of TLR2 is significantly lower in lane 4 than lane 1; however TLR2 densitometry figures showed identical TLR2 densitometry in these lanes. The authors should explain how the densitometry was calculated and normalized. It appears that the main difference is in the untreated lanes.*

We have added more detail regarding our densitometry calculation to the figure legend and the methods. Briefly, we calculated densitometry through ImageJ and normalized to hour 0 for each condition. Lane 4 (hour 0 for both Ubiquitin overexpression and PPP1R11 overexpression) has lower TLR2 protein level to start due to the inherent enhanced derogatory environment of additional Ubiquitin and additional E3 ligase.

*2) In Figure 2, it appears that TLR2 degradation started from 1h following Pam3CSK4 stimulation, whereas PPP1R11 binding did not occur until 4h. Pam3CSK-induced TLR2 ubiquitination also started at 1h post-stimulation (Figure 2). The authors should repeat this experiment to see whether this result is reproducible or due to experimental variations. If these data are reproducible, they would suggest the existence of another E3 that mediates the early stage of TLR2 ubiquitination. In such a case, the authors should discuss the results.*

We have repeated this experiment and observed increased association of TLR2 and PPP1r11 at 1-hour post-stimulation (Figure 2). This is consistent with the data in Figure 2, in which TLR2 ubiquitination level increased at 1h following Pam3CSK4 treatment.

*3) The mechanism underlying TLR2 ubiquitination was not investigated in depth. The authors should use chain-specific ubiquitin antibodies to examine whether TLR2 is conjugated with K48 ubiquitin chains or other chains (e.g. Figure 1 and Figure 2).*

This is an excellent point. We immunoprecipitated TLR2 and observed significant K48-linked ubiquitin signal (Figure 1—figure supplement 1). This is consistent with our data from the UBICrest assay, which suggests K48-ubiquitin linkages on TLR2. However, this does not rule out other species of ubiquitin linkages not associated with the degradation of TLR2. While study of these linkages will further help elucidate TLR signaling, we believe it is beyond the scope of this manuscript.

*4) In Figure 1, the protein level of TLR2 is drastically reduced after incubating TLR2 with E1, E2 and PPP1R11 in* in vitro *ubiquitin conjugation assays. Is this due to ubiquitination or degradation in the* in vitro *system? This result is quite different from that of Figure 1, central panel (MG-132).*

The protein level is due to degradation in the system. The in vitro ubiquitination assay composition includes rabbit reticulocyte lysate as part of the protein synthesis process in the TnT kit. In the past we have noticed baseline degradation due to the presence of proteasome and other degradative mechanisms in the reticulocyte lysate. We also hypothesize accelerated degradation occurs due to the cell-free nature of the assay.

*5) Figure 1 should indicate the amount of the individual DUBs and show a panel of DUB protein immunoblot.*

Thank you for this comment; it is important as high concentrations of DUBs can lead to promiscuity. The working concentrations of the DUBs were 0.5 μM, except for OTUD7A which was 0.1 μM. This experimental detail has been added to the methods section. Additionally, we included a Coomassie gel to show the presence of DUB proteins in the reactions (Figure 1—figure supplement 1).

*6) In Figure 1—figure supplement 3, the authors find several key residues of PPP1R11 that are important for TLR2 degradation. It would be helpful if the authors the possible structural mechanisms and any rationales for testing these residues (e.g. are these conserved residues in certain domains?).*

We have included a cartoon showing the non-canonical ring finger domain including cluster of residues we mutated in the study (Figure 1—figure supplement 3). We believe these residues form a putative set of atypical RING domains, granting PPP1R11 activity as an E3 Ligase. Further, we included a negative control mutation (H126A) which retains activity in TLR2 degradation (Figure 1—figure supplement 3).

*Reviewer #2:*

*[…] First, most of the data in this manuscript are from ectopic expression or RNAi-mediated gene knockdown experiments. As gene-specific knockout using the CRISPR-Cas9 system is technically mature, data collected with those PPP1R11 knockout cell-lines or mice, if applicable, should undoubtedly make the conclusion more solid and reliable.*

We used CRISPR/Cas9 system to generate PPP1R11 knockout in MLE cells and have evaluated the KO cells in response to TLR2 ligand Pam3CSK4 in vitro. Utilizing sgRNAs designed by Santa Cruz Biotechnologies, we generated a 2 bp deletion in the first exon of PPP1R11 in MLE cells leading to an immediate pre-mature stop codon. PPP1R11 KO cells also show significantly prolonged TLR2 half-life, relative to control cells, and an increase cytokine release following ligand treatment (Figure 2—figure supplement 2). This result is highly consistent with the PPP1R11 siRNA knockdown experiments (Figure 2,5F–G).

*Second, the effects of PPP1R11 in TLR2-mediated infection are double faced. From the gene transfer experiments, we can see that the gain or loss of PPP1R11 results in a selection between a control of inflammation and bacteria clearance. This phenomenon, however, makes it a hard decision to manipulate this gene for an effective treatment of lung infection. The authors should carefully think over this issue and determine the physiological effects of PPP1R11.*

*Third, while the manuscript is well written, there are areas that would benefit from re-working in terms of English grammar and usage. These would make the manuscript flow better and, therefore, make it more pleasant for the reader. I would like to stress however that, in its current form, the manuscript is both readable and understandable.*

*Specific Points:*

*1) The control panel probing TLR2 from Figure 1 does not show a time-dependent decrease in the protein level of TLR2, for the TLR2 level at 4 hour treatment is higher than that at 2 hour treatment. This inconsistency should be corrected.*

This most likely is due to experimental variations from immunoblotting. We do however showed the time-dependent decrease of TLR2 with Pam3CSK4 treatment in additional three experiments Figure 2. Nevertheless, we reran the samples and included new immunoblots that display the time-dependent decrease in TLR2 protein in the control condition (Figure 1). We believe these results suggest ligand-induced TLR2 protein degradation occurs in a time-dependent manner, and is blocked by MG132 treatment.

*2) Figure 1, another RING finger E3 ligase should be included in these experiments side by side with PPP1R11 to show that the latter is specific for TLR2. Using PCGF1 as in Figure 1 is just fine.*

We have conducted CHX chase experiments utilizing PCGF1 over-expression and siRNA silencing in MLE cells and observed no change in TLR2 protein decay relative to control (Figure 1—figure supplement 2). We also calculated and plotted protein densitometry. PCGF1 is RING-domain E3 ligase, and hypothesized to function similar to PPP1R11. These experiments illustrate the specificity in the E3-substrate relationship.

*3) Figure 2, an siRNA for another RING finger E3 ligase should be included in these experiments to show specificity. Using Trim52 as in Figure 2 is just fine.*

We silenced Trim52 in MLE cells followed by Pam3CSK4 treatment, and measured IL-6 and CXCL-1 cytokine release through ELISA (Figure 2—figure supplement 1). Similar to experiments with PCGF1(Figure 1—figure supplement 2), we observe that a fellow RING E3 Ligase is unable to affect a functional phenotype related to PPP1R11 in regard to TLR2 degradation and cytokine release. While we do not claim PPP1R11 to be the sole-regulator of TLR2 degradation, it has proven sufficient to degrade TLR2 in contrast to fellow RING E3 family members.

*4) Figure 3, although a general trend is shown that PPP1R11 expression results in the loss of TLR2, individual consistency should be paid attention to, as gene transfer Mouse 1 should not display more TLR2 than Mouse 5 considering their PPP1R11 expression levels.*

We have processed frozen tissue samples and re-immunoblotted for TLR2 and PPP1R11 resulting in more representative and more consistent protein signal (Figure 3). There exists inherent variability among individual mice as to their baseline protein concentrations and the effect size of their response to challenge. We believe that fresh sample processing and western blot displays better consistency among replicates.

5) Figure 4, similar to the point raised for Figure 3, and more consistent panels are preferred for PPP1R11 and TLR2 expressions, if applicable.

We have processed frozen tissue samples and re-immunoblotted for TLR2 and PPP1R11 resulting in more representative and more consistent protein signal (Figure 4). There exists inherent variability among individual mice as to their baseline protein concentrations and the effect size of their response to challenge. We believe that fresh sample processing and western blot displays better consistency among replicates.